# A Bio-Inspired Endogenous Attention-Based Architecture for a Social Robot

**DOI:** 10.3390/s22145248

**Published:** 2022-07-13

**Authors:** Sara Marques-Villarroya, Jose Carlos Castillo, Juan José Gamboa-Montero, Javier Sevilla-Salcedo, Miguel Angel Salichs

**Affiliations:** RoboticsLab, Universidad Carlos III de Madrid, 28911 Leganés, Spain; jgamboa@ing.uc3m.es (J.J.G.-M.); javier.sevilla@uc3m.es (J.S.-S.); salichs@ing.uc3m.es (M.A.S.)

**Keywords:** perception, social robots, bio-inspired attention, human–robot interaction

## Abstract

A robust perception system is crucial for natural human–robot interaction. An essential capability of these systems is to provide a rich representation of the robot’s environment, typically using multiple sensory sources. Moreover, this information allows the robot to react to both external stimuli and user responses. The novel contribution of this paper is the development of a perception architecture, which was based on the bio-inspired concept of *endogenous attention* being integrated into a real social robot. In this paper, the architecture is defined at a theoretical level to provide insights into the underlying bio-inspired mechanisms and at a practical level to integrate and test the architecture within the complete architecture of a robot. We also defined mechanisms to establish the most salient stimulus for the detection or task in question. Furthermore, the attention-based architecture uses information from the robot’s decision-making system to produce user responses and robot decisions. Finally, this paper also presents the preliminary test results from the integration of this architecture into a real social robot.

## 1. Introduction

In human–robot interaction (HRI), it is essential to select the most relevant stimuli to achieve a natural experience. In some cases, this interaction encounters constraints from the perception capabilities of the robot, especially considering the computational resources that are needed by state-of-the-art perception techniques, such as those that are based on deep learning. Therefore, to create an agile interaction with a high level of detail, a compromise is required regarding the number of detectors that can run simultaneously, in most cases. Detectors are associated with the number of stimuli and the quality and delay of the detections. Consequently, mechanisms that select the most salient stimuli should play an essential role in HRI; however, these selection mechanisms are often omitted. These mechanisms are similar to those in animals and are related to a key concept: attention.

There are different definitions of the term *attention*. Talsma et al. defined it as a multisensory cognitive function that allows humans and animals to continuously and dynamically select a particular stimulus from all of the available information in their environment [1]. Similarly, Broadbent characterised attention as the selective filtering of input stimuli to make the amount of data to be processed more manageable [2]. The common concept is that attention is a mechanism for selecting the most relevant stimulus in the environment. This selection means that some detectors (e.g., selectors that are associated with specific tasks) can be active only when required and can otherwise be idle or run with fewer resources when the interaction does not require them.

Typically, attention is a multisensory process, although vision is the most studied modality. According to Stein et al., the multisensory response occurs when inputs from different modalities (i.e., sight, touch and hearing) elicit a single response [3]. This result is usually amplified when the inputs appear in the same space and are synchronised in time, meaning that single-sensor stimuli are weaker.

Another interesting concept is the *focus of attention* (FOA). The FOA is the stimulus that was selected by the attentional mechanism and is then processed in detail. Information from regions outside of the FOA is either stored in the short-term memory or, in most cases, ignored, which causes a phenomenon called *change blindness* in the peripheral areas [4]. This event improves perceptual efficiency by only processing the most relevant information from the environment and allowing an increased processing and resolution load to be focused on the FOA and not on stimuli within the periphery [1].

When selecting the most relevant stimulus in a scene, we have to consider that humans detect objects within a visual location better when we know some of the characteristics beforehand (colour, movement, etc.). This type of known information is called the *perceptual set* [5]. According to Rosenbaum et al., the response time to stimuli is shorter when the user knows what kind of movement to expect in advance. This information is known as the *motor set* [6]. The perceptual and motor sets together comprise the *attentional set*, which is defined as the representations that are involved in selecting the relevant stimuli and responses to perform a task correctly.

The selection of the FOA can be either endogenous or exogenous. *Endogenous attention*, also called top-down or voluntary attention, is goal-driven and directed towards events or stimuli that are consciously selected by the individual [7]. This attention can optimise performance according to the task demands and may be maintained at a location for extended periods. In contrast, *exogenous attention*, also called bottom-up or involuntary attention, is driven by the importance of the stimuli and is an adaptive tool that enables the detection and processing of salient events that appear outside of the FOA. Exogenous attention can be understood as a momentary interruption of endogenous attention or the reorientation of attention towards a different stimulus. According to Corbetta et al., involuntary attention mechanisms, although primarily stimulus-driven, are modulated by goal-directed influences through attentional groups as they impose task completion as the priority measure [8]. Therefore, most attention-grabbing stimuli are related to the task at hand [9]. To ensure that a social robot can handle multiple stimuli while focusing on its current task, a multisensory endogenous attention-based system is necessary to filter out irrelevant stimuli. In addition, these systems allow for better resource management because they give priority to the detectors that are associated with the current task.

The main contribution of this work is the definition and development of a novel perception architecture that integrates bio-inspired concepts from biological attention and focuses on endogenous components. Our multisensory architecture sorts the detected stimuli in order of importance, considering the tasks that are performed by a social robot and focusing on HRI since the architecture is proposed for use in a social robot. We extend the existing endogenous attention systems by integrating stimuli from different modalities (vision, sound and touch) into a single system. In addition, we introduce concepts such as sustained and punctual attention to prioritise HRI and achieve natural interactions. Finally, we tested our system using different applications of a real robotic platform and achieved competitive responses in real time.

The rest of this article is structured as follows. Section 2 offers a selection of the most relevant works within the field of attention and studies the main mechanisms that govern these processes. Artificial attention models are also explored to consider their advantages and limitations and the most common types of sensory sources. Section 3 introduces the robotic platform that was employed in this study and the integration of our software architecture. Section 4 presents the main contribution of this work: the bio-inspired perception architecture that was based on endogenous attention. Section 5 introduces the experimental methodology that was followed to assess the proposed architecture. Section 6 presents the main results, which are organised into three case studies, and Section 7 analyses these results and explores the limitations of the system in its current state. Finally, the main conclusions that were drawn from this work are presented in Section 8.

## 2. Background

The selection of predominant stimuli is an innate mechanism in the animal world that offers significant advantages for survival, for example, hearing sounds that indicate danger or particular patterns in the environment [10]. Attention is, therefore, a multisensory process that involves many sources of information. Animals complement visual clues with those from other senses, such as hearing, smell or touch. The following sections review the mechanisms that drive the attention processes in living organisms, focusing on those that are related to endogenous attention.

### 2.1. Natural Attention Models

Several studies have highlighted the different mechanisms that underlie the attention process. For example, when selecting the most relevant features of a scene, one of the most influential human attention theories is *feature integration theory* [11]. This theory suggests that our perception of the environment occurs in two phases: an early stage (pre-attention), in which the individual features of an object (colour, shape, motion, etc.) are processed, and a late phase, which focuses on finding the FOA by fusing those particular features into a single conjunction of properties (i.e., an object). The *Simon effect* measures the response time to visual stimuli and demonstrates that the position of a stimulus is directly related to our ability to react to it [12]. Herbranson explored the hierarchical components within attention mechanisms, through which users can shift from focusing on a specific feature to a global analysis of the situation [10].

In some cases, humans can divide attention and allow different relevant stimuli to be focused on simultaneously, although accuracy is reduced compared to analysing features separately. For example, Cherry et al. studied the *cocktail party effect* phenomenon using a listening-based task [13]. Participants listened to two conversations at the same time. Punctual attention allowed them to focus on one of the two conversations while filtering out the other, so this type of attention allows the ability to focus a specific stimulus or activity in the presence of other distracting stimuli. In this experiment, the authors demonstrated that it was impossible to focus on both conversations simultaneously. This phenomenon also occurs with visual stimuli [14].

Endogenous attention voluntarily processes task-related stimuli and features within the environment. In the literature, works such as [15,16,17] have divided this attention mechanism into sustained attention and punctual attention (which is also referred to as selective attention in the literature). Sohlberg et al. defined sustained attention as the ability to maintain a consistent behavioural response during continuous or repetitive activity and punctual attention as the ability to maintain a cognitive set that requires the activation and inhibition of responses, depending on the discrimination of stimuli. Fisher described sustained attention as the ability to maintain sensitivity to incoming stimuli over time and selective attention as the ability to process part of the sensory input while excluding others.

Shulman et al. showed that these processes occur in specific regions of the brain [18]. For example, areas that are sensitive to movement react poorly to changes in colour. Therefore, we can establish a parallelism between certain regions in the brain that are specialised for certain detection tasks and detectors in software that tend to process a single kind of stimuli or detections of the same nature (e.g., an object detector is specialised for that kind of stimuli and is blind to movement). A change in task also usually causes a modification in the neural response. For example, changing from sorting numbers (odd and even) to sorting letters (vowels and consonants) produces a momentary decrease in the performance of the second task, which is known as *change cost* [19].

### 2.2. Artificial Attention Models

Over the years, there has been an increase in interest around the design of artificial attention models. These models select the most relevant parts of the environment and filter out irrelevant information to enable the faster processing of the important stimuli [20,21].

The field of robotics has significantly benefited from these systems as they have allowed robots to become more autonomous. Therefore, mechanisms that are capable of selecting relevant and helpful information are essential for robotics applications, especially when processing high-resolution and high-frequency sensory information, such as images.

Computational attention models usually process the different stimuli in the environment according to the task that is to be performed [22]. These models can react to a stimulus or task in a manner that is comparable to humans or other animals. The processing in such models can be divided into three phases [23]:*Orientation* selects where or what feature to focus on next (for example, in a white room with a red dot on a wall, that dot would be the relevant stimuli);*Selection* establishes how to focus on the feature (for example, the system chooses whether to pay attention to auditory or visual stimuli);*Amplification* decides how to process the selected stimulus, which is different from how it processes non-relevant features (for example, when we see a red dot in a white room, amplification selects the best method to analyse that object but in the rest of the room, object recognition may not be used).

Over recent years, there have been many examples of artificial attention systems in the literature, which have mainly focused on vision. Along this line, Meibodi et al. [24] presented an attention model that uses exogenous attention, endogenous information and memory to complete the attention process. Another example is the work of Zhu et al. [25], who predicted the head and eye movements of their participants using attention studies. In that work, they incorporated concepts such as visual uncertainty and balance to achieve good results in their predictions. Another example that is worth mentioning is the work that was developed by Yang et al. [26], who presented an inverse reinforcement learning model to predict the next endogenous focus of attention in humans. Along the same line, Fang et al. [27] revised the role of top-down modelling in salient object detection and designed a novel densely nested top-down flow (DNTDF)-based framework. They compared their results to other datasets to verify the correctness of their model. Finally, Adeli et al. [28] proposed an attention system that was based on using neural networks to predict the next focus of attention in real scenarios. They used bio-inspired concepts, such as the ventral, frontal and visual areas of the human brain.

#### Endogenous Attention Models

The literature offers different approaches for endogenous models; although in most cases, these are descriptive and are not supported by concrete developments [23]. This has been caused by a lack of generality in the algorithms. For example, it is crucial to evaluate the placement of an object within a room to understand the relevance of that object for a given task. Therefore, generic object recognition algorithms are not valid for these models [29]. Nevertheless, we can find approaches that implement attention models, such as the work of Wang and Shen [30], and use exogenous information to predict endogenous FOAs in complex situations. Another important concept is feature-based attention. This allows for the precise location of the property of interest in time, e.g., identifying upward movements when the task is to control a lifting movement [31]. In these models, the effects of endogenous knowledge indicate how exogenous stimuli should be processed.

Theses endogenous feature-based attention models emerged first [32,33]. Soon after, architectures such as the *guided search theory* [34] and the *feature gate model* [35] appeared. These architectures attempt to answer how feature gains should be adapted to achieve optimal task performance. All of these models have a supervised learning phase to compute property gains using manually labelled examples [36]. A common characteristic of all of the works that have been presented in this section is that they tend to output a region in the selected space (or image) as the following FOA. Based on these models, Beuter et al. [37] proposed fusing the exogenous and endogenous information within an image to determine the most salient areas in order to control a robot’s navigation. In the same way, Yu et al. [38] presented an artificial visual attention manager that uses the features of an object to guide the perception of the robot to the next FOA, according to the current task, context and learned knowledge.

A more recent example that uses probabilistic reasoning and inference tools is the model that was proposed by Borji and Itti [39]. This work introduced an architecture for modelling endogenous visual attention that was based on reasoning in a task-dependent manner. Their model analyses the semantic values of objects within a scene and tracks the user’s eye movements, then combines both with a probabilistic approach to predict the next object that is needed by the user to perform their task.

The works that have been presented in this section use endogenous attention as a complement to exogenous information and place more importance on salient stimuli that are related to the task that is to be performed. However, to the extent of our knowledge, no work in the literature has proposed a purely endogenous architecture that reacts in real time. Moreover, works on attention management that is applied to robotics are scarce and they have usually focused on mobile robots that need attention to navigate correctly. In contrast, this paper presents an endogenous attention manager for social robots that prioritises natural human–robot interaction over other stimuli and real-time reactions. This system analyses stimuli from different sources (sight, sound and touch) to decide which is the most relevant stimulus, taking into account all of the information in the environment (not just visual information as in the literature). This feature allows the robot to interact with its environment and perform more complex tasks that include multimodal detectors.

## 3. The Robotic Platform

To test our approach, we used the social robot Mini, which utilises the perception architecture that was presented in Salichs et al. [40]. Mini was developed at RoboticsLab and is a desktop robot that is 59 cm tall and can share its emotional state through expressive eyes, an LED heartbeat, cheeks and the movement of its arms, head and base. The aim of this robot is to assist and entertain older adults who suffer from mild cognitive problems. In terms of the robot’s hardware, Mini has LEDs in its heart and cheeks, a VU meter as a mouth and two OLED screens that serve as eyes. Between the head and the body, Mini has a neck with two degrees of freedom that allows head movements in the vertical and horizontal axes. It also has two arms with one degree of freedom and a base, which are all controlled by servomotors. These elements allow Mini to achieve a natural vivacity. Mini also includes capacitive sensors to detect tactile stimuli and an external tablet to enhance its interaction with users by displaying multimedia content, such as videos, images or buttons, for games and exercises. The base holds a computer that has an Arduino microcontroller and a small battery for safe shutdown. Finally, the robot has an RGB-D camera at its base (Realsense D435i camera: https://www.intelrealsense.com/depth-camera-d435i/ (accessed on 10 July 2022), see Figure 1). For particular skills (e.g., to play tangram), we installed a USB fisheye camera (USBFHD04H 1080P H.264 camera: http://www.elpcctv.com/elp-free-driver-1080p-full-hd-h264-usb-webcam-camera-module-for-car-bus-plane-video-surveillance-p-350.html (accessed on 10 July 2022)) with a 180${}^{\circ }$ field of view, which provided images with a resolution of 1080p at 30 frames per second, on the robot’s chest. That camera focused on the table between the robot and the user to see the game pieces.

### 3.1. The Software Architecture

The robot’s software architecture is divided into blocks, as shown in Figure 2. Our work focused on the attention manager and detectors that run the robot. The detector and actuator blocks are directly connected to the physical devices of the robot and acquire information from the sensors or by commanding the actuators, respectively.

The *skills* correspond to the main functionalities of the robot and each skill uses perceptual information to achieve its goal. The skills also use HRI capabilities to generate dialogues and interact with the user. The robot integrates various categories of skills, such as the following:**Games** contains all of the games that Mini can play, such as bingo, akinator [41], tangram [42] and quiz games, which were chosen to stimulate elderly people following the recommendations of doctors or psychologists;**Multimedia** includes all of the multimedia capabilities that the robot can use (e.g., showing pictures or videos, playing music, audiobooks or movies and telling jokes);**Information** contains the information that the robot can provide, such as the news or weather [43], as well as an introduction for itself and instructions on how to interact with it;**Cognitive stimulation** incorporates all of the cognitive stimulation exercises (e.g., memory, attention, planning and comprehension exercises);**General** contains the essential communication elements for the user and the robot, such as reminders or relevant information data;**Sleep** includes the robot’s configuration to simulate sleep (i.e., eyes closed and a neutral position of the arms, base and head).

The *human–robot interaction* (HRI) system manages the dialogue between the user and the robot. This part of the architecture processes and analyses perceptual information to decide whether it is relevant to the interaction; for example, when a skill needs the robot to provide information or the robot is waiting for an answer from the user. The robot only communicates with the user using voice or the menus on the tablet, but it also uses visual cues on the tablet and different gestures to improve communication. Finally, the *decision-making system* (DMS) controls the activation and deactivation of the robot’s functionalities. This component uses the robot’s internal information (e.g., motivation) and environment information (e.g., the user requesting a specific skill) to proactively decide the robot’s next action.

To test the operation of the attention manager, we explored three case studies. In the first case study, we used a multimedia skill, specifically the telling jokes skill (Section 5.2). Next, we tested a cognitive stimulation skill, specifically a memory game that is based on recognising famous monuments around the world (Section 5.3). Finally, we tested a game skill, specifically the tangram game, by adding vision detectors to the attention manager (Section 5.4).

## 4. The Attention Manager

This paper proposes a bio-inspired endogenous artificial attention manager for social robots. We took inspiration from psychological and neuroscientific works that considered how human attention works within social interactions. It is important to note that the main purpose of social robots is to provide human–robot interaction as naturally as possible. To identify the requirements for developing this bio-inspired system, we studied how endogenous attention works in biological systems and its application to HRI, which led us to choose the following specifications for the attention manager:To decide on the optimal FOA considering the available stimuli and the current task;To process information from multiple sensory modalities (e.g., vision, touch and sound);To respond in real time, avoiding lengthy processing and response times;To select the correct FOA to achieve natural human–robot interaction;To include a voluntary attention process that detects the relevant information for a goal in a sustained manner;To include a mechanism that allows the robot to focus on a stimulus punctually, even when it is not the most salient stimulus, in a sustained manner;To take into account the current task of the robot (i.e., the system needs to communicate with the DMS to know what task the robot is currently performing and its next action and then, with this information, the system can sort the stimuli according to their relevance to the task);To filter out irrelevant stimuli for the current task;To consider scalability and modularity (i.e., allow the inclusion of new sources of information);To manage resources efficiently (i.e., inhibit or disable detectors depending on the task).

According to these requirements, we developed the architecture of the bio-inspired endogenous attention manager, as shown in Figure 3. In the following sections, we explain each of the blocks that make up the proposed architecture in more detail.

### 4.1. The Detectors

The detectors lie at the base of the architecture. They are responsible for extracting information from the robot’s environment and generating a rich representation. To optimise the computational load of the robot, the architecture can activate the detectors only when they are necessary.

According to their modality, we classified the detectors into visual, auditory and tactile. This modality is related to how the attention manager analyses the specific relative importance of each stimulus, as explained in Section 4.2. Additionally, the detectors could be organised by type: interaction, endogenous and mixed.

Mini uses *interaction detectors* to achieve bidirectional communication with the user. In our case, the robot had two types of interaction detectors: automatic speech recognition (ASR) [44] for voice communication and the tablet [45], which the user used for tactile feedback. As explained in Section 3.1, the HRI manager controls the robot’s dialogue. Therefore, it handles both the questions that are asked by the robot and the answers that are given by the user. The attention manager receives the user’s answers from the detectors and through which channel they were produced using the information from this software module (see [46] for more details on how the HRI manager works in the Mini). The interaction detections are then positioned in the space around the robot. In the case of the tablet, we assumed that the device was always in a fixed place in front of the robot. When the robot uses the ASR, the attention manager tries to locate the user’s face to select it as the FOA. Our architecture integrated face detection and localisation system called face-detection-retail-004, which relies on Squeezenet and an SSD network (face detector network: https://docs.openvinotoolkit.org/latest/_models_intel_face_detection_retail_0004_description_face_detection_retail_0004.html (accessed on 10 July 2022)). This detector worked well with a frontal view of the user’s face, achieving an average accuracy of 83%. For each detection, the algorithm provides a unique user ID, the confidence level for the predicted class and the coordinates of the upper left and the lower right corners of the face’s bounding box.The attention manager uses this information to fix the FOA in the centre of the detection box when the robot has to pay attention to the user (for example, when the robot is talking or waiting for an answer from the user).In contrast, when the user is not sitting in front of the robot, the attention manager uses the Mini’s omnidirectional microphone to locate the direction of the user’s voice. For this, the architecture uses the Open embeddeD audition System (ODAS) library because it allows for the 3D localisation, tracking, separation and filtering of sound sources [47].Additionally, when the robot is waiting for a voice response from the user and it does not detect anybody, the attention manager tries to find the direction of the sound to locate the user, even when the user is behind the robot. Conversely, when the face detection system finds a user, the system prioritises the face over the voice stimuli.*Endogenous detectors* are specifically developed to provide information for a certain skill. Therefore, the robot only activates these sensors when it needs them to perform the relevant skill.The attention manager fully controls the activation and deactivation of these detectors. Focusing on a specific skill that was used in the case studies, the tangram game skill integrates three endogenous vision-based detectors: the first is in charge of detecting the play zone, the second is responsible for performing the calibration that is necessary to play correctly and the third is used to recognise and locate the game pieces. We assumed that the game board was in a fixed position between the user and the robot to place these stimuli. This assumption meant that all of the detectors that were operating in the playing area had the same FOA, which was the centre of the board.*Mixed detectors* can detect endogenous and exogenous information in the environment. For example, considering the face detector, when the robot is idle and a person enters the room, the face that is detected captures the robot’s attention exogenously. In contrast, when the robot is waiting for a user response during a game, the user’s face captures the robot’s attention endogenously. These two behaviours result in the detector being considered as a mixed detector. These detectors are always active, but the attention manager only uses their data when they are required for the current skill. Apart from the face detector, the Mini system uses other three mixed detectors: the first detects tactile interactions, the second locates the user’s face and the third recognises the direction of sound. To detect and identify tactile events, the robot uses an algorithm that is capable of detecting touch in three areas (the belly and the left and right arms) using capacitive sensors. When the attention manager detects a tactile event, the attention focuses on the relative position of the sensor that was activated with respect to the centre of the robot.

### 4.2. Updating Detector Information

Endogenous attention is a voluntary process; therefore, it is directed by the task that the robot is performing. To add this feature into the attention manager, the system needs to communicate with the DMS that is in charge of deciding which task the robot performs at a given time. In our case, the DMS informed the attention manager when there was a change in the robot’s skill and whether the change involved the activation of a new ability or the deactivation of the current skill. This communication has to be asynchronous and the architecture needs to adapt quickly to the requirements of the new skill. When the robot initiates a new skill, the attention manager uses the skill detectors database (see the *Skills Detectors Database* in Figure 3) to load the associated information, such as the detectors that are associated with that skill and their relative importance (weight) in terms of attention.

As stated at the beginning of Section 4, the architecture has to process information from different sensory modalities. Our robot could react to different visual, auditive and tactile stimuli and each sensory modality had a relative importance that depended on the task that the robot was performing and whether the stimulus was relevant to the HRI. However, these modalities do not represent the same information. In our case, the stimuli were different, so we did not need a fusion step to merge information from different sensors that were detecting similar stimuli at the same location.

Visual stimuli were associated with the task itself and were not used by the HRI mechanisms. For example, when the robot played tangram, the detector returned the position of the pieces on the playing area. Equation (Equation 1) characterised the weight or importance of this type of detector:(1)$$\omega_{vision}=\left\{\begin{array}{ccc} 1.0, & if & n_{vision\;detectors}>0\\ 0.0, & otherwise \end{array}\right.$$

Voice interaction was also considered. For example, when the robot asked a question and waited for an answer. Considering the importance of HRI for these robots, sound stimuli were more critical than visual stimuli and, therefore, had a higher relative importance value (see Equation (Equation 2)):(2)$$\omega_{audio}=\left\{\begin{array}{ccc} 2.0, & if & n_{audio\;detectors}>0\\ 0.0, & otherwise \end{array}\right.$$

In our robot, tactile stimuli were considered as an interruption to the task. For example, when the robot told a joke, the user could touch its belly to stop the activity. Similarly, when the robot was playing the tangram game, the user could touch its arm to request a hint. Since different studies have demonstrated that humans pay more attention to tactile stimuli, the attention manager placed more importance on these detections. Equation (Equation 3) defined the weight of this type of stimuli:(3)$$\omega_{tactile}=\left\{\begin{array}{ccc} 3.0, & if & n_{tactile\;detectors}>0\\ 0.0, & otherwise \end{array}\right.$$

This system not only focuses on events that continuously appeared over time (sustained attention), but it also reacts to punctual stimuli at specific times (punctual attention). In this last case, a stimulus may not be as salient as the sustained a priori stimuli, but to calculate their relevance, the temporal factor has to also be considered. This mechanism is especially useful when the robot maintains a dialogue with the user via different channels (e.g., voice or tablet). In these cases, sustained attention tends to pay attention to the user’s face, but the system must correct this issue to allow for variations between all of the possible stimuli. For this reason, the attention manager introduces a correction factor to the sustained and selective stimuli to modify their weights at each time step. This correction factor multiplies the relative importance of a specific stimulus by a factor of three. This way, the robot is able to focus its attention on the response channel with the correction factor.

The attention manager communicates with the HRI system, which controls the dialogues, and informs the general architecture about which input method the user selected to respond (see the upper right box in Figure 3).

In the current version of the attention manager, we assumed that the stimuli that could trigger punctual attention just came from the interaction detectors, specifically the automatic speech recognition and the tablet detector. Equation (Equation 4) shows how the system calculated the final importance of the interaction detectors:(4)$$\left(\begin{array}{c} \omega_{FinalASR}\\ \omega_{FinalTablet} \end{array}\right)=\left(\begin{array}{c} \omega_{ASR}\\ \omega_{Tablet} \end{array}\right)*\left(\begin{array}{cc} \omega_{SA1} & 0.0\\ 0.0 & \omega_{SA2} \end{array}\right)$$
where $\omega_{ASR}$ and $\omega_{Tablet}$ are the a priori weight of the detectors at each time step and $\omega_{SA1}$ and $\omega_{SA2}$ are the correction factors that corresponded to punctual attention for the two detectors (the $\omega_{SA1}$ and $\omega_{SA2}$ values were always either 1.0 or 3.0, depending on the detector’s attention type (sustained or punctual); for example, when the detector produced punctual attention because the user interacted with the robot using that channel, the correction factor was 3, in the other cases, the value was 1).

### 4.3. Creation of the Ego-Spheric Representation

The system requires the integration of stimuli from different sensory sources. In human attention, the brain performs a sensory integration process that allows us to perceive information coherently [1]. Our architecture integrates this process and enables the generation of a unique representation for all endogenous stimuli.

In our work, we used an ego-spheric representation to achieve multisensory aggregation. We took inspiration from the work of Bodiroza et al. [48] and the representation allowed the rendering of any stimuli around the robot, not only those in front of the camera. The sphere represented a multimodal egocentric map, in which the system recreated the areas of the salient stimuli that were located by the robot. In the ego-spheric representation, the robot’s centre corresponded to the sphere’s centre and the system showed the salient data as small spheres around the Mini. The size of each small sphere in the representation depended on the importance of that stimulus. Moreover, the position of each stimulus was relative to the centre of the robot. Equation (Equation 5) was used to calculate the final saliency map, where ω is the relative importance depending on the sensory modality that was tuned-up (as shown in Section 4.2), *F* is the required detector, *d* is the total modalities and *c* is the final combined map:(5)$$c_{t}=\sum_{j=1}^{d}\omega_{jt}\;\cdot \;F_{jt}$$
$$\begin{array}{cc} \mathrm{where}\;\;c_{t} & =\mathrm{is}\;\mathrm{the}\;\mathrm{final}\;\mathrm{combined}\;\mathrm{map}\;\mathrm{in}\;\mathrm{instant}\;t\;\\ j\; & \epsilon \;\{1,2,...,number\;of\;detectors\;activated\}\\ \omega_{jt} & =\;is\;the\;relative\;importance\\ F_{jt} & =\;is\;the\;needed\;detector\\ d\; & =is\;the\;total\;modalities \end{array}$$

The classical approach of saliency maps is to only represent the stimuli in front of the camera and to not take depth into account [23]. Therefore, they usually only include visual stimuli. However, in our work, we included multisensory information in the representation that could appear at any place around the robot, including tactile, auditive and visual stimuli. In the Figure 4, the yellow mark represents the centre of the robot and the pink sphere is the space around it. Note that the green dot is bigger than the others, meaning that the stimulus was more significant than the others.

In our architecture, the most salient stimulus is not the only output. Instead, the endogenous attention processing generates a sorted list that includes all of the stimuli that exceed the salience threshold.

## 5. Methodology

In Section 3.1, we explained that the Mini can perform a wide range of skills. In this section, we describe how the attention manager works and how it exchanges information with the other software architecture blocks using three realistic case studies. The first case study involved the system performing an entertainment skill: joke telling. The second case study evaluated the attention manager when performing a cognitive stimulation skill, more precisely a memory and attention exercise that consisted of recognising famous world monuments. Finally, the third case study showed how the system works when performing a game skill: the tangram game. For each of these skills, we describe how the different blocks of the software architecture communicated during the execution of the skill and the response of the endogenous attention manager. We focused on where the robot placed the FOA and the differences between its sustained and punctual attention. Table 1 summarises the case studies that are presented in this work. It indicates the type of skill, the objective of the test and the types of detectors that were involved. The objectives were cumulative, i.e., in case study 2, we aimed to meet the goals of case study 1 as well as those of case study 2.

### 5.1. Experimental Setup

In the case studies, the user sat in front of the robot (see Figure 5). In this case, four kinds of sensors provided the information about the robot’s environment: an omnidirectional microphone *ReSpeaker Mic Array v2.0* (ReSpeaker Mic Array v2.0: https://wiki.seeedstudio.com/ReSpeaker_Mic_Array_v2.0/ (accessed on 10 July 2022)), an RGB-D camera (Realsense D435i: https://www.intelrealsense.com/depth-camera-d435i/ (accessed on 10 July 2022)), three capacitive touch sensors that were placed on the arms and belly of the robot and a tablet (Samsung Galaxy Tab A: https://www.samsung.com/es/tablets/galaxy-tab-a/galaxy-tab-a-10-1-inch-white-32gb-lte-sm-t585nzwephe/ (accessed on 10 July 2022)), which was placed between the robot and the user.

As well as from the sensors, the robot integrated a GPU USB *Intel Neural Compute Stick* (Intel NCS2: https://ark.intel.com/content/www/us/en/ark/products/140109/intel-neural-compute-stick-2.html (accessed on 10 July 2022)) to extend the robot’s processing capabilities. Finally, the third case study included additional accessories for the skill, such as a play zone, which was also placed between the robot and the user, and a set of tangram pieces. All of the software modules were connected using ROS [49] and the system operated at 3 fps for visual stimuli, which still allowed the robot to have a human-like reaction time [50] while controlling the computational load.

To test the functioning of the system, the research team performed ten repetitions of each case study and the times that are presented in Section 6 are those that were obtained in the last repetition. In addition, the system was stressed by performing the different skills consecutively and triggering all of the possible options in each skill that was selected for the case studies. The average interaction time was 80.3 s in case study 1, 269.1 s in case study 2 and 148.7 s in case study 3.

### 5.2. Case Study 1: Attention during an Entertainment Skill

In this case study, we used an entertainment skill that tells jokes to the user [43]. The user could choose the type and subject of the jokes. The execution was divided into two states. The first state was selection of the skill, for which we assumed that the robot was initially awake and wanted to interact with the user. Then, the robot asked the user what they wanted to do using different communication channels (voice and the tablet). The user answered using one of these channels. After the selection, the joke telling skill initiated and the robot told three jokes to the user using voice communication.

In this case, we checked the system response when several detectors were active simultaneously and the different results that were obtained depending on whether the attention was sustained or punctual. Another important aspect was the communication between our system and the other software blocks within the robot (the HRI system, DMS and skills). Finally, the case study also checked the correct functioning of the output stimuli list, which was sorted according to salience. Section 6.1 shows the step-by-step operation of the Mini software and the behaviour of the attention manager during the case study in detail.

### 5.3. Case Study 2: Attention during a Cognitive Stimulation Skill

This case study involved a cognitive stimulation exercise that was based on memory and attention [51]. During the exercise, the Mini displayed a well-known monument, such as the Eiffel Tower or the Coliseum, and asked the user for their relative city, giving three choices.

As in the previous case study, there were two stages. The first stage consisted of the selection of the skill. To test a different set of inputs, in this case, the user asked the robot for the exercise directly by saying “I want to play to the monuments game”.

In the second part of the case study, we evaluated the operation of the attention manager while the user was completing the stimulation exercise. During the execution, the Mini requested information using its text-to-speech functionality and activated the ASR as the default communication channel. However, when the ASR detected a communication problem (recognition failure or no response from the user), the Mini continued the communication using the tablet as the input. The game had seven different questions and for the first three, the attention manager received the user voice as the input, while for the last four questions, it acquired information from the tablet.

In this case study, we tested the attention manager with a more complex skill and in a dynamic setting, in which we intended to check that robot produced the correct output when interacting with the user and when changing communication channels. Section 6.2 presents the results from the communication between the software modules in the Mini and the attention manager output during the exercise.

### 5.4. Case Study 3: Attention during a Game

In this case, the user played the tangram game and the robot controlled the game development and helped the user by providing hints. In this case, the robot used an additional camera. This device focused on the table between the robot and the user to detect the play zone of the game, in which the user could freely move and place the tangram pieces (see Figure 5). To simplify the case study, this case did not include the selection state as it was similar to those of the previous case studies. During the game, the robot used endogenous detectors that were explicitly developed for this skill, such as the play zone detector, calibration detector or tangram piece detector, as described in [42]. Furthermore, the robot also included touch detection to provide hints to the user. Finally, the robot asked the user questions using the different communication channels (voice and the tablet) during the game.

For this skill, the robot used multiple kinds of detectors as it needed endogenous, interaction and mixed detectors to function correctly for the game. Therefore, in this case study, we tested the system attention output in a complex scenario using stimuli from different sensory modalities (sight, sound and touch). Section 6.3 details the connections between the different software blocks and the output of the endogenous attention manager.

## 6. Results

This section presents the results that were obtained by the complete endogenous attention manager in the different types of robot activities. Based on the attention manager specifications that were described in Section 4, the obtained results helped to verify that the attention manager satisfied the following requirements:Able to identify the most salient stimulus for a given task;Able to handle multimodal stimuli (visual, auditory and tactile);Real-time responses;Able to select the most relevant information to achieve natural human–robot interaction;Able to react to sustained and punctual attention;The correct management of the activation and deactivation of detectors to reduce computational load.

### 6.1. Case Study 1: Attention during an Entertainment Skill

In this case study, we considered that the user wanted to interact with the robot and that they wanted to listen to short, robot-related jokes. Figure 6 shows a sequence diagram with the messages that were exchanged among the different Mini software modules during the execution of the joke telling skill. Additionally, Figure 7 summarises the results of this case study, showing the relative importance of each detector during sustained attention and punctual attention.

This case study consisted of two parts: the selection state and the jokes state. Initially, the robot was awake and initiated the interaction by asking the user what they wanted to do (at t=0 s). This question caused the attention manager to activate the available communication channels, initiating the tablet and ASR detectors. At this point, the system paid attention to the user’s face and the tablet in a sustained manner.

As explained in Section 4.1, when the robot expected a voice response, it also tried to locate the face of the closest user, assuming that this user was the person who responded. Furthermore, using the weights that were discussed in Section 4.2, the attention manager placed more importance to the user’s face, despite both detectors being active.

After a few seconds, the user answered using the tablet; therefore, there was a punctual change in the attention manager to place more importance on the device for a few seconds (see the blue point in Figure 6 and the red point in Figure 7 at t=13 s). Next, the robot prompted three more questions (from t=14 s to t=16 s, from t=17 s to t=19 s and from t=20 s to t=28 s in Figure 6) to select which entertainment skill the user wanted and which topic they liked jokes about. The output of the attention manager for these three questions was identical to that described for the first question: the user answered using the tablet.

Once the user selected the skill, the case study continued in the jokes state (at t=29 s). This state described the behaviour of the system during the execution of the joke telling skill. This skill was the simplest among the three case studies since the robot only interacted with the user through voice. Note that when the robot was talking, the sustained attention manager located the user’s face to look at this point when the Mini was talking to achieve a natural interaction. Therefore, the Mini paid attention to the user’s face while telling each joke and in the goodbye message (from t=28 s to t=35 s, from t=38 s to t=57 s, from t=59 s to t=67 s and from t=68 s to t=77 s in Figure 7).

### 6.2. Case Study 2: Attention during a Cognitive Stimulation Skill

In this case study, we considered that the user wanted to perform a specific cognitive stimulation exercise: the monuments game. In this game, the robot displayed pictures of famous monuments on the tablet and asked the user for their relative city, giving three possible answers. This exercise had seven different questions and during the case study, the user answered the first three questions using voice. During the fourth question, a communication error appeared and forced a change of communication channel and the user responded to the last four questions using the tablet.

The Figure 8 shows a sequence diagram with the messages that were passing among the different Mini software modules during the case study. Complementarily, Figure 9 summarises the results of this test, showing the relative importance of each detector during sustained attention and the points at which there was a punctual change in the robot’s attention.

As in case study 1, this case study was divided into two parts: the selection state and the exercise state. The robot was initially awake and started the interaction by asking the user what they wanted to do (at t=0 s). In this case, the user selected a cognitive stimulation exercise using voice (at t=11 s in Figure 9). The behaviour during this first part of the case study (selection state) was similar to that in case study 1. When the robot asked a question, the attention manager activated the communication channels to receive the user’s answer (ASR and tablet). Despite having both detectors active, the sustained attention placed more importance on the user’s face due to the weights that were discussed previously. In this case, since the user answered using the voice, the most relevant stimulus for the attention manager corresponded to the punctual attention (see the red point in Figure 9).

Once the user selected the skill (at t=20 s in Figure 8), the robot briefly introduced the exercise and explained what the user had to do. During this speech, the attention manager activated the face detector to pay attention to the user (from t=22 s to t=44 s in Figure 9). Then, the Mini asked for the Eiffel Tower’s location, giving different options for the answer (Paris, Berlin or Rome). In this case, the user could reply using ASR, so the attention manager activated this channel (at t=45 s) and the user responded after a few seconds (at t=65 s).

As this was a correct answer, the robot responded with positive feedback for the user. During this interaction, the attention manager activated the face detector again because the user was the most salient stimuli (from t=66 s to t=76 s in Figure 8). The game continued in this way for two more questions (from t=77 s to t=90 s and from t=100 s to t=123 s in Figure 9). Next, the robot asked where the Alhambra is, so the attention manager again activated the ASR (at t=133 s in Figure 8). When a user does not answer within a predetermined time or the speech recognition fails, the detector reports a communication error between the robot and the user. In this case, the robot prompted the user to use the tablet to respond (at t=207 s in Figure 8). As in the previous case, the attention manager activated the face detector to look at the user while talking. Then, the robot repeated the question; However, in this case, it displayed the options on the tablet so that the user could answer. At this point, the attention manager activated the tablet’s detector (from t=218 s to t=229 s in Figure 9) and waited for the user’s response (see the red point in Figure 9 at t=229 s). Finally, when the robot received the answer, the attention manager activated the face detector again to provide the feedback to the user (from t=230 s to t=240 s in Figure 8).

In summary, the exercise had seven questions and the user used voice to answer the first three (see the red points in Figure 9 at t=65 s, t=90 s and t=123 s). During the fourth question, we simulated a communication problem with the ASR (from t=133 s to t=205 s) and the user responded using the tablet (see the red point at t=229 s in Figure 9). The user also answered the last three questions using the tablet (see the red points at t=250 s and t=269 s in Figure 9).

### 6.3. Case Study 3: Attention during a Game

This case study tested the operation of the attention manager when a user played the tangram game with the robot. In this case, the user played with a physical tangram and the robot controlled the development of the game using computer vision to detect the play zone and pieces and helped the user when they needed help. This last case was the most complex of the three case studies, as it included seven detectors and more complex interactions. As in the previous cases, Figure 10 shows a sequence diagram describing the messages that were passed among the different Mini software blocks during the execution of the game and Figure 11 presents the relative importance of each detector during sustained and punctual attention.

In this case study, we omitted the game selection phase to focus on how the FOA evolved during a complex game. The dynamics of this discarded phase were identical to those of the previous case studies. Therefore, the robot started the interaction by welcoming the user to the game. During the welcome message, the attention manager activated the face detector because the robot was talking to the user, so the most salient stimulus was the user’s face (t=0 s to t=9 s in Figure 11). Next, the robot explored the area in front of it to find the play zone (t=10 s to t=14 s in Figure 10). When the robot could not see the play zone, the Mini informed the user that the game could not start. The next step involved a calibration to enhance the tangram detector accuracy; hence, the robot provided some instructions to the user (t=15 s to t=35 s in Figure 11).

During this time, the robot focused its attention on the user’s face. Before starting the calibration process, the robot asked the user to confirm that they had followed the instructions (at t=38 s in Figure 11). In the same way as in the cognitive stimulation case study, when the user did not respond or the speech recognition failed (see the red point in Figure 11 at t=90 s), the Mini repeated the question but displayed the options on the tablet (at t=91 s in Figure 10). Due to the change in the communication channel, the attention manager deactivated the ASR and activated the tablet detector. When the user responded (see the red point in the Figure 11 at t=103 s), the skill started the calibration process (at t=113 s in Figure 11). The attention manager activated the calibration detector and waited until the calibration process was successful (at t=114 s in Figure 10). As with the play zone detection, when the robot could not perform the calibration, the Mini informed the user that it could not play at that moment and ended the game.

Once the calibration was completed, the robot was ready to play and provided instructions to the user on how to play the game (at t=115 s in Figure 11). At that moment, the attention manager activated the face detector and a few seconds later (at t=132 s in Figure 11), it activated the detector that recognises tangram pieces. Although two sustained attention detectors were active simultaneously (face and tangram pieces), there was no punctual attention in this case, so the robot always paid more attention to the user’s face than to the board due to the preset weights ($\omega_{audio}$ = 2.0 and $\omega_{vision}$ = 1.0). When the Mini finished giving the instructions, the attention manager deactivated the face detector (at t=136 s in Figure 11) and the robot started paying more attention to the pieces on the board.

Additionally, during the tangram game, the user could touch the robot at any time to ask for a hint. In this case study, that occurred at t=143 s in Figure 10. At this point, the attention manager deactivated the tangram pieces detector and activated the face detector to look at the user while giving a clue (at t=144 s in Figure 11). When the robot finished the clue, it switched its attention from the user to the game pieces (at t=157 in Figure 11).

## 7. Discussion

In this study, we tested whether an endogenous attention system allowed our robot to identify the most salient task-related stimulus, regardless of the modality (sight, sound or touch), in real time. In addition, we verified whether the robot could prioritise interaction-related stimuli to achieve the most natural HRI possible using bio-inspired techniques, such as sustained and punctual attention. Finally, enabling the robot to manage the activation and deactivation of the detectors was a priority to reduce the computational load and produce the correct functioning of the robot. We tested these requirements in three different scenarios, all of which proved satisfactory results.

In the first two case studies, we tested the system’s ability to identify the most salient stimulus to perform the given task during the selection stage. In this part of the case study, two possible stimuli stood out in a sustained way: the ASR and the tablet. In the first case study, although the ASR was more salient in a sustained way, the tablet was more relevant when the user answered the questions using the tablet and not using their voice. We tested whether the proposed attention system could identify the most relevant stimulus when there was more than one possible task-related stimulus in the environment and whether it could react correctly to sustained and punctual attention. Moreover, in these two case studies, we also checked that the robot could correctly control the activation and deactivation of the detectors, both in the selection phase and in the execution of the skill itself. This reduced the computational load of the system and ensured the correct function of our social robot.

On the other hand, in case study 2, we verified that the system could adapt to a change in communication channel due to an error in the voice recognition system. This adaptation showed the system’s adaptability in real time as this change did not cause any delays in the attention system’s response and there were no additional problems due to the error during the interaction.

Focusing on case study 3, it is instantly noticeable that this case study was more complex than the previous cases. Firstly, we tested how the system reacted to stimuli from three sensory modalities (sight, sound and touch). The results from this case study showed that the robot had an excellent adaptation to the different sensory modalities and that their detection and localisation did not cause any delays in the system. Furthermore, in this case, we also verified whether the system could prioritise the stimuli that were necessary for the interaction over the stimuli that were related to the task itself. This prioritisation allowed the robot to interact with the user in the most natural way possible as it prioritised human–robot interaction.

During the development of the tests, we researched other endogenous attention models in the literature to compare to our system. However, we did not find any multimodal implementations that could allow this comparison. Furthermore, many of the models that have been presented in the literature are not purely endogenous, but rather use exogenous information to guide endogenous attention. Another problem was the lack of multimodal attention systems, including visual, auditory and tactile stimuli. Therefore, we focused on performing different tests on our system to check that it was working correctly.

Despite the advantages of the proposed system, there were also some limitations. Firstly, the integration of detectors for the stimuli of different modalities allowed us to make the robot capable of performing more complex tasks that included visual, auditable and tactile stimuli. However, the system was unable to analyse whether stimuli from different sensory modalities belonged to the same stimulus (e.g., detecting a person and their associated voice at the same time). We hope to explore this in future work by including a fusion level that merges stimuli from different modalities and relates them to each other.

On the other hand, although the results that were obtained in the case studies were satisfactory, using preset weights for the relative importance of each sensory modality did not allow the robot to adapt to its task. Furthermore, these preset values meant that the robot always reacted in the same way when performing a task, which could lead to errors on certain occasions. Therefore, in future work, we hope to include a machine learning level that directly adjusts these weights using deep learning techniques. This automatic learning would take into account past experiences and current activity so we could able to customise the system to the user that it interacts with and the robot’s tasks.

Finally, the case studies allowed us to check the correct functioning of the system during different skills. The research team conducted the tests in 2021. Unfortunately, due to the current COVID-19 situation, we could not test this system with elderly people. Moreover, although the implementation phase is complete, the system needs to be validated in a controlled environment before long-term testing with real users. Despite this, we hope to perform tests with the target population to check the correct functioning of the system. In addition, we plan to carry out long-term tests that would allow us to check that the system responds correctly in stressful situations.

## 8. Conclusions

This work presented a bio-inspired endogenous attention architecture for social robots. The architecture aims to detect the most relevant stimulus in a given environment and consider the necessary adaptions for the robot’s behaviour in order to complete the current task. Moreover, the system’s reaction time aims to be similar to that of a human for the same process. Therefore, the model aims to work in real time. The system also considers stimuli from various sensory modalities. The ego-spheric representation provides a 3D view of the stimuli within the robot’s environment.

Our experimental results showed that our system met the requirements that were set out in Section 4. Firstly, we developed a communication process with the DMS that allowed the robot to know the current task at all times. Moreover, as the results show, the adaptation of the attention manager was fast and always less than one second, thus avoiding possible bottlenecks that could be produced by our architecture. In the same way, the attention manager could process information from different sensory modalities, including visual, auditory and tactile detectors. We added a detector block to the software that allows for the individual preprocessing of environment information and works in parallel to avoid delays in the system.

The results also indicated that the system could work with both sustained and punctual attention in the three case studies. The system correctly sorted the stimuli according to salience using the preset weights and modified them when there was a punctual stimulus. In addition, with the previously fixed relative importance values, the attention manager selected the most important stimuli for a more natural human–robot interaction in a sustained way.

A machine learning level would dynamically adjust these weights in future work by considering past experiences and current activity. Moreover, a fusion level would merge the stimuli from different modalities to allow the robot to understand the relationships between the detections. Finally, we also hope to perform long-term tests with elderly people to check the functioning of the system and its adaptation to a stressful environment.

A critical architecture requirement was real-time responses. The case studies demonstrated that the stimulus localisation and saliency classification worked in less than one second in all cases.

As for the system’s modularity, we verified that the architecture also satisfied this requirement using different algorithms. In the first two case studies, the system only used interaction detectors, while in the last case study, we added endogenous and mixed detectors and we achieved the correct output in all cases. These results also demonstrated that the system could activate and deactivate the detectors when necessary, thus efficiently managing its resources.

## Figures and Tables

**Figure 1 sensors-22-05248-f001:**
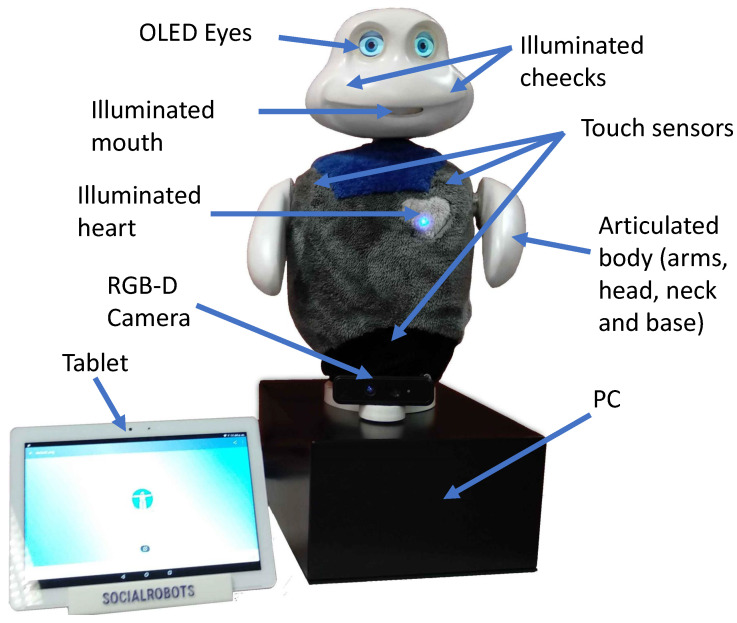
The social robot Mini.

**Figure 2 sensors-22-05248-f002:**
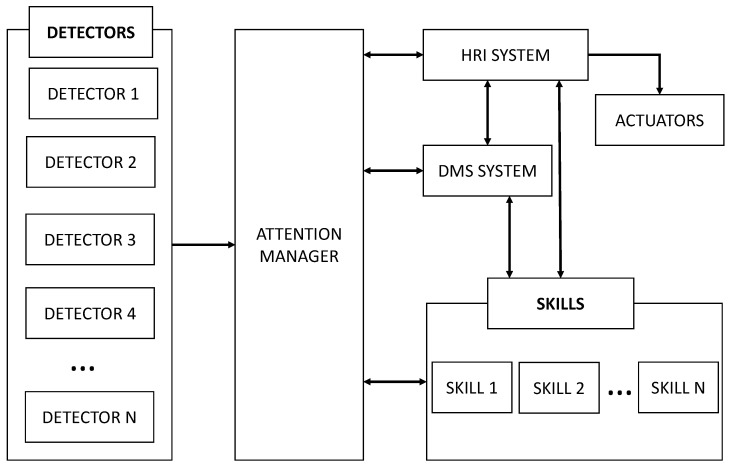
A general overview of the main software components of the Mini architecture and the connections between them. This work added the attention manager block, which communicated with the HRI and DMS systems and the robot’s skills.

**Figure 3 sensors-22-05248-f003:**
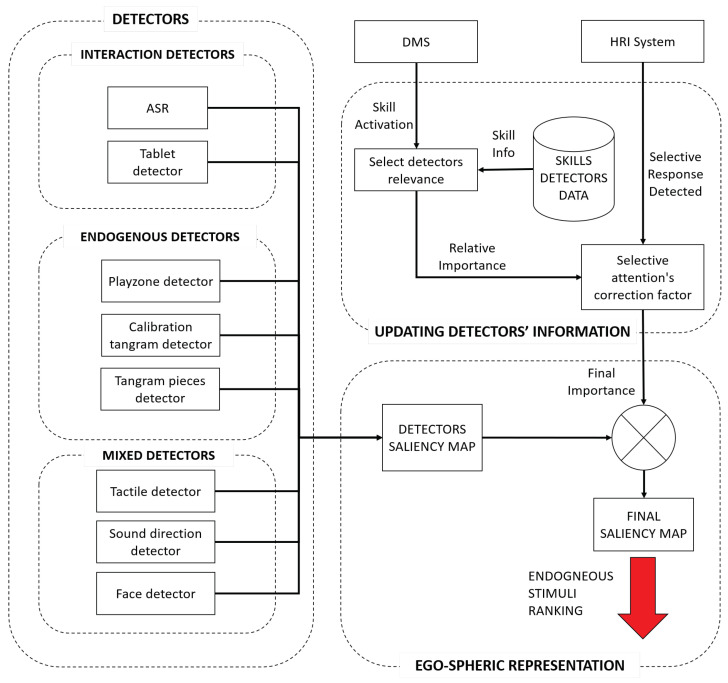
A schematic diagram of the endogenous attention manager.

**Figure 4 sensors-22-05248-f004:**
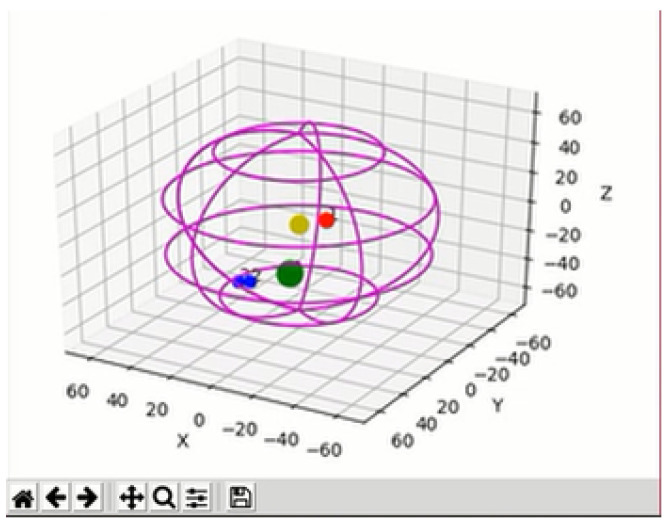
An example of an ego-spheric representation with simultaneous tactile (green circle), auditive (red circle) and visual (blue circle) stimuli.

**Figure 5 sensors-22-05248-f005:**
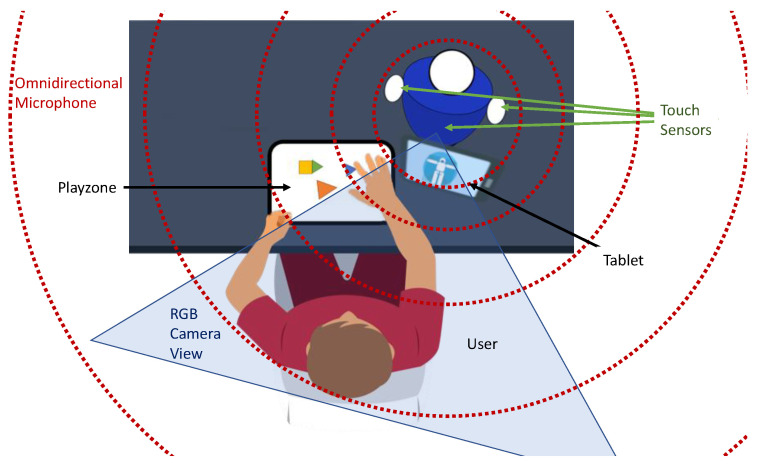
An illustration of the environmental setup of the case studies. We used the same colour coding as that in the 3D representation of the environment (blue for the camera, green for the touch sensors and red for the speaker).

**Figure 6 sensors-22-05248-f006:**
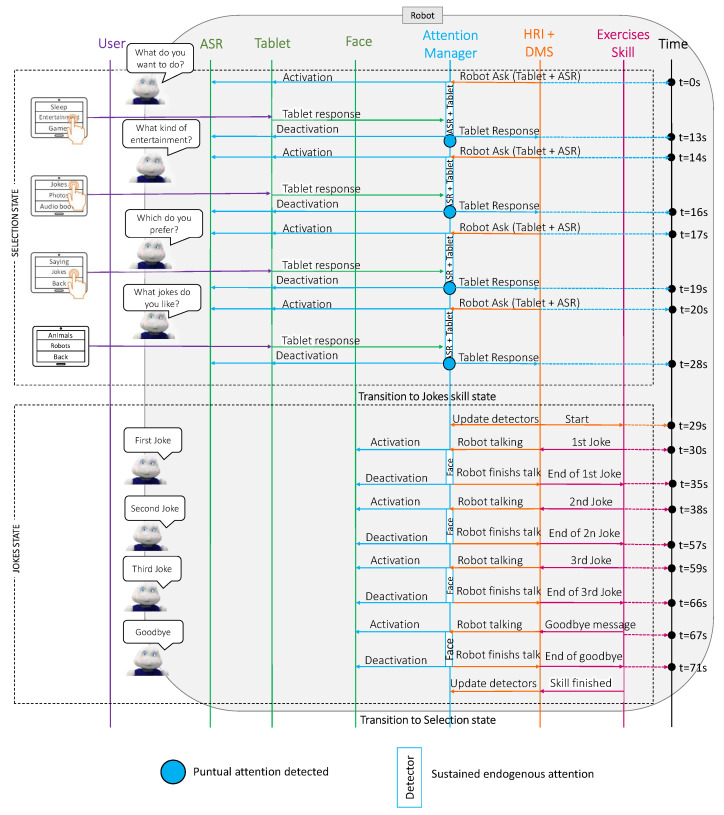
A sequence diagram of the connections between the different software blocks in the Mini during the execution of the joke telling skill. The blue boxes in the attention manager row display sustained attention from the detectors. The blue dots show the punctual attention that was produced by the user’s responses.

**Figure 7 sensors-22-05248-f007:**
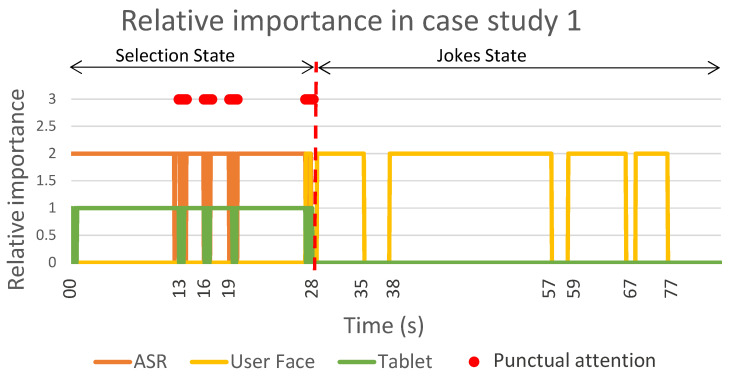
The relative importance of the detectors during sustained and punctual attention in case study 1. The orange, yellow and green lines show the sustained attention of the ASR, the user’s face and the tablet, respectively. The red dots display the punctual attention that was detected during the case study.

**Figure 8 sensors-22-05248-f008:**
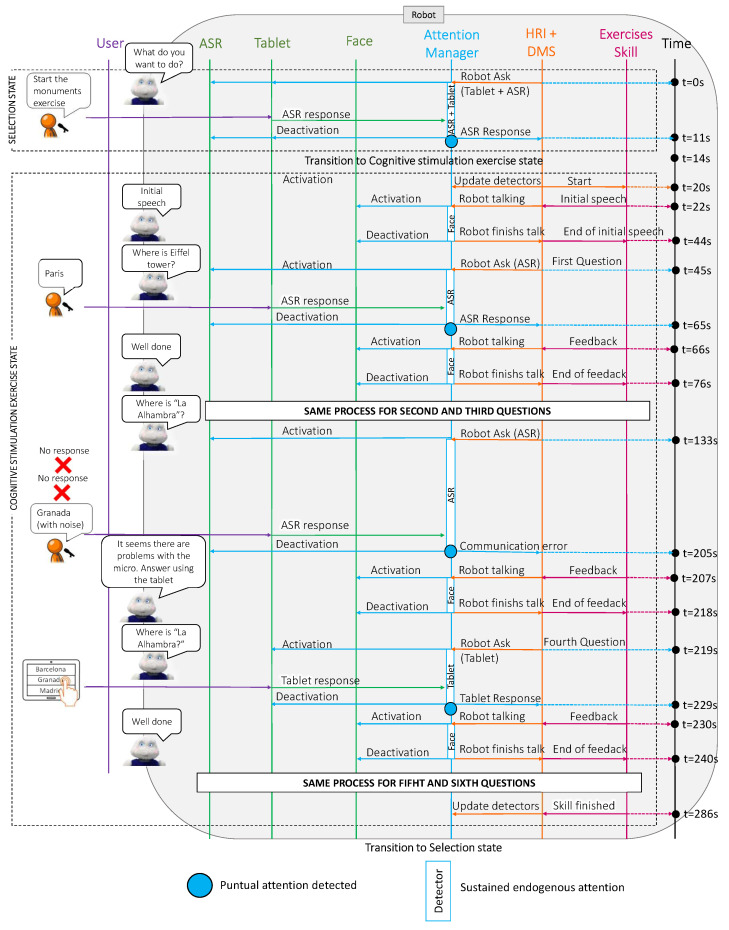
A sequence diagram of the connections between the different software blocks in the Mini during the execution of the cognitive stimulation exercise.

**Figure 9 sensors-22-05248-f009:**
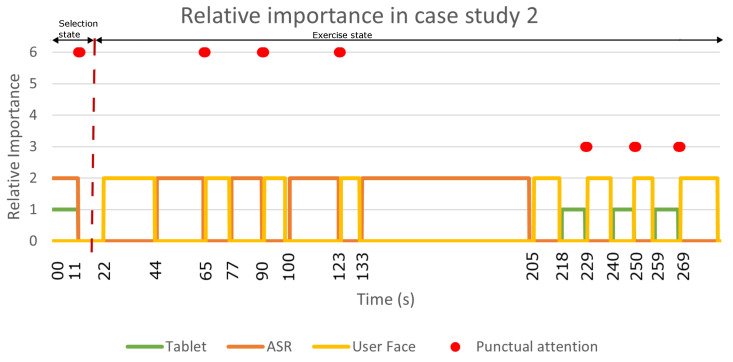
The relative importance of the detectors during sustained and punctual attention in case study 2. The orange, yellow and green lines show the sustained attention of the ASR, the user’s face and the tablet, respectively. The red dots display the punctual attention that was detected during the case study. The user’s answers produced the first three dots using ASR, while the tablet responses produced the last four dots.

**Figure 10 sensors-22-05248-f010:**
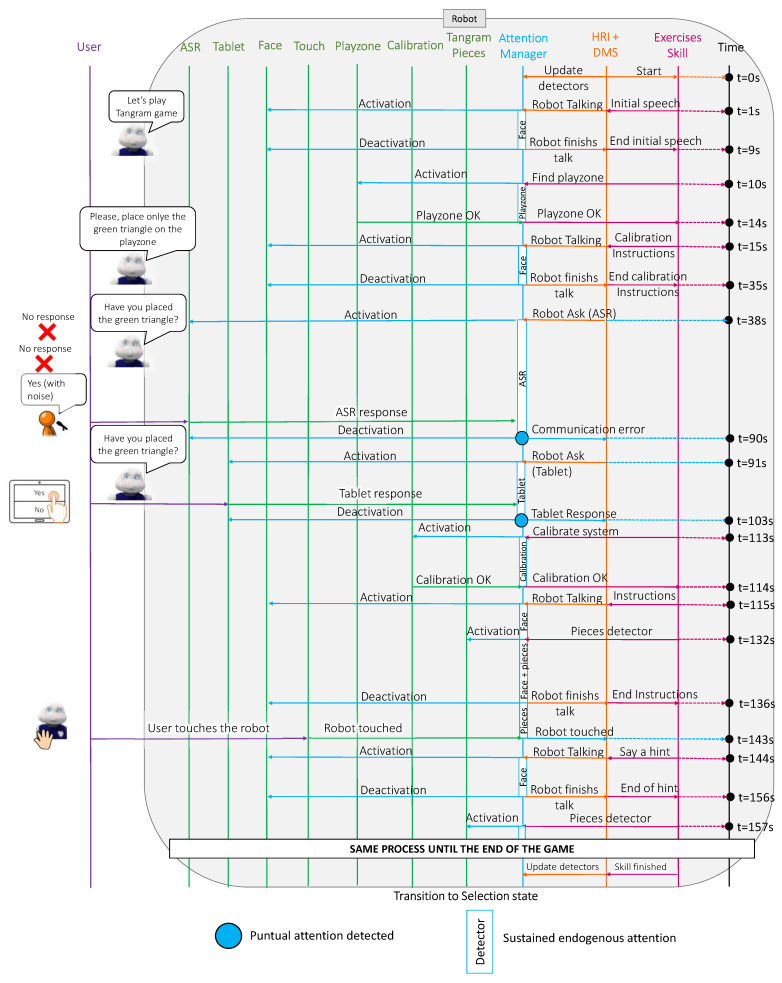
A sequence diagram of the connections between the different software blocks in the Mini during the execution of the tangram game.

**Figure 11 sensors-22-05248-f011:**
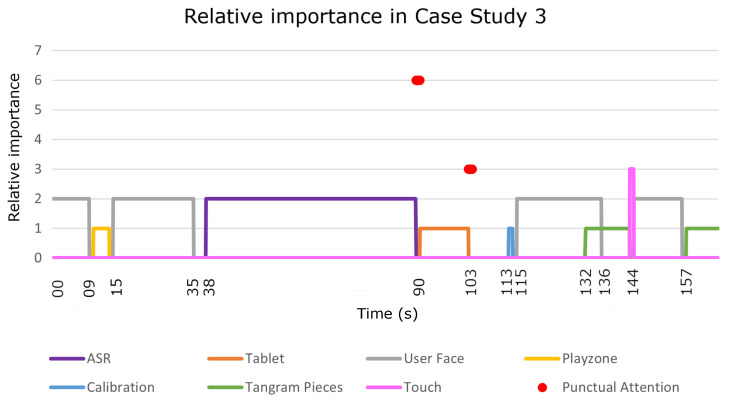
The relative importance of the detectors during sustained and punctual attention in case study 3. The lines show the sustained attention of the different detectors that were used during the case study. The red dots show the punctual attention that was detected. The first dot was produced by the user’s response via ASR and the second dot was produced by the user’s response via the tablet.

**Table 1 sensors-22-05248-t001:** A summary of the three case studies that are presented in this work.

	Skill	Goal	Detectors Involved
Case Study 1	Entertainment	- System responds to multiple activated detectors - System responds to sustained or punctual attention- Communication with other software blocks - Sorted list according to salience as the output	- Interaction detectors - Mixed detectors
Case Study 2	Cognitive Stimulation	- System responds to changes in communication channels	- Interaction detectors - Mixed detectors
Case Study 3	Game	- System responds to stimuli from different sensory modalities	- Interaction detectors - Mixed detectors - Endogenous detectors

## Data Availability

Not applicable.

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
