# Peer review of "A Bio-Inspired Endogenous Attention-Based Architecture for a Social Robot"

_sensors, 2022, doi:10.3390/s22145248_

Round 1

Reviewer 1 Report

It is a well written paper, and explains the findings of the research very well. One minor comment would be to present statistical datasets on repeatability of the system response. Mention the number of times you performed these tests and whether the response to stimuli was stress tested. 

Reviewer 2 Report

it's not clear face detector is different face recognition? why?  is there two cameras? , is there priority for each detector?.

It's not clear the difference(functionalities) between detector and recognition. 

Authors mention that "the architecture can process information from different sensory modalities", so the architecture how process information multimodal, where different sensors can represent a same information. On another hand, the robot can communicate/express through  different channels  as voice, visual,  text and gesture??. 

Case studies, How long was the user's interaction with the robot, what type of user was he/she? The model is focused on older adults with cognitive problems.

Reviewer 3 Report

This paper introduced endogenous attention architecture for social robots, which aims to detect the most relevant stimulus in the environment and consider that the robot’s behavior has to adapt to the current task. The system also considers that stimuli from various sensory modalities may appear. Overall, the paper is interesting and well organized, however, in the reviewer's opinion, the manuscript requires revision, in order to address the following concerns,

1.     The novelty of the paper should be addressed in the Introduction section clear and simply.

2.     How to determine and adjust the parameters in Equation 5.

3.     Is there any shortcoming of the developed framework? The authors are suggested to make some discussions.

4.     The authors may propose some interesting problems as future work in the conclusion.

This study may be proposed for publication if it is addressed in the specified problems.

Reviewer 4 Report

I enjoyed reading this paper that follows a very practical approach and tries to measure the results of the approach considering three scenarios. The approach is correct but there are some shortcomings in terms of a greater scientific framework and greater detail and methodological rigor.

Improvement suggestions:

- I suggest the authors to improve the abstract to clarify the innovative level of this study and to provide an overview of the implications for theory and practice.

- The information given regarding artificial attention models is very superficial. I consider that there is space for significant improvement and for the inclusion of more current literature references as this is a topic that has evolved strongly in the last 5 years.

- Authors state “Mini was developed at the RoboticsLab and is a small desktop robot (0.59 m. heigh).”. I think that the readers would be interesting to know more about the physical capabilities of this robot. Authors could provide more information regarding it.

- Authors note: “Finally, the robot has an RGB-D 2 camera placed at the base (see Figure 1)”. Does it have only a single camera? Why its positioning on the base?

- Figure 2 gives the wrong idea that only 3 skills are considering. I think it would be important to revise the image.

- Authors state “Games: contains all the games that Mini can play, such as Bingo, Akinator, Tangram or quiz games.” What were the criteria for choosing these games?

- I was a little disappointed with Figure 2. It is a very high level image that does not give technical information with great accuracy. I would like to better understand the physical and logical architecture of the solution. Perhaps splitting the physical and logical architecture into two images would be preferable.

- The 1st paragraph of the methodology section is very important. However, it is not easy to understand the differences and objectives of each case study. I think it would be desirable to present a summary table with this information.

- It would be desirable to have more information about the context in which the case studies were conducted. It is not clear on what date it was carried out and who plays the role of user, namely whether it involved users external to the research team.

- One of the main shortcomings of this study is the discussion of the results. It lacks a comparative discussion of the results obtained in each case study and also to compare the results obtained with other solutions proposed in the literature.

- As future work, authors only state “In future work, a machine learning level will dynamically adjust these weights taking into account past experiences and current activity.” It is crucial to give more precise information. 

- Authors should explicitly address the limitations of their study.

- Authors should also address the main theoretical and practical contributions of their proposed solution.

Round 2

Reviewer 4 Report

The review work done by the authors was very positive and well supported.